# Suspicions of two bridgehead invasions of *Xylella fastidiosa* subsp. *multiplex* in France

Enora Dupas [1,2], Karine Durand [1], Adrien Rieux[3], Martial Briand [1], Olivier Pruvost[3], Amandine Cunty [2], Nicolas Denancé[1], Cécile Donnadieu[4], Bruno Legendre[2], Céline Lopez-Roques[4], Sophie Cesbron [1], Virginie Ravigné[5,6] & Marie-Agnès Jacques [1,6 ✉]

Of American origin, a wide diversity of *Xylella fastidiosa* strains belonging to different subspecies have been reported in Europe since 2013 and its discovery in Italian olive groves. Strains from the subspecies *multiplex* (ST6 and ST7) were first identified in France in 2015 in urban and natural areas. To trace back the most probable scenario of introduction in France, the molecular evolution rate of this subspecies was estimated at $3.2165 \times 10^{-7}$ substitutions per site per year, based on heterochronous genome sequences collected worldwide. This rate allowed the dating of the divergence between French and American strains in 1987 for ST6 and in 1971 for ST7. The development of a new VNTR-13 scheme allowed tracing the spread of the bacterium in France, hypothesizing an American origin. Our results suggest that both sequence types were initially introduced and spread in Provence-Alpes-Côte d'Azur (PACA); then they were introduced in Corsica in two waves from the PACA bridgehead populations.

[1] Univ Angers, Institut Agro, INRAE, IRHS, SFR QUASAV, F-49000 Angers, France. [2] French Agency for Food, Environmental and Occupational Health & Safety, Plant Health Laboratory, Angers, France. [3] CIRAD, UMR PVBMT, F-97410 Saint Pierre, La Réunion, France. [4] INRAE, US 1426, GeT-PlaGe, Genotoul, Castanet-Tolosan, France. [5] CIRAD, UMR PHIM, F-34398 Montpellier, France. [6] These authors contributed equally: Virginie Ravigné, Marie-Agnès Jacques. ✉email: marie-agnes.jacques@inrae.fr

Worldwide global food security is threatened by emerging plant diseases, which frequency increases as a consequence of the globalization of plant material exchanges, crop intensification and global climate change[1]. Many emerging infectious diseases result from geographical expansion following accidental introduction and can be considered as biological invasions[2]. Our ability to anticipate, prevent and mitigate emerging infectious diseases depends on a better understanding of current pathogen distribution over space and time, invasion routes, conditions favoring their emergence and population reservoirs[3].

*Xylella fastidiosa* is a phytopathogenic genetically diverse bacterial species including two largely divergent lineages. One lineage bears two subspecies, *fastidiosa* and *multiplex*, originating from Central and North America, respectively[4,5]. The other lineage is composed of one highly diverse subspecies, *pauca*, known to originate from South America[6]. To type *X. fastidiosa* at an infrasubspecific level, MultiLocus Sequence Typing (MLST), which consists in partial sequencing of seven housekeeping genes and leads to sequence type (ST) number attribution, is the reference method used[7].

In Europe, since the first detection of *X. fastidiosa* made in Italy in 2013[8], the bacterium has devastated thousands of hectares of olive groves in Apulia, leading to a severe socio-economic crisis[9]. *X. fastidiosa* has a very wide host range, with 664 known host plant species worldwide[10]. Its host range includes plants of major socio-economic interest such as grapevine, citrus, coffee and olive trees, but also fruit and forest trees, ornamental plants, shade trees, and wild species, making *X. fastidiosa* a global threat and ranking it among the most dangerous plant-pathogenic bacteria worldwide[11]. For these reasons, *X. fastidiosa* is a priority quarantine pathogen in the European Union (EU), meaning that it is of mandatory declaration and eradication, with the exception of areas where the bacterium is established, and where eradication has been considered as unfeasible[12,13]. In EU, these areas are Corsica in France, Apulia in Italy, and the Balearic Islands in Spain[12,14].

Over short distances, the dissemination of the bacterium mostly results from its only natural way of transmission among plants, the movements of *Hemiptera* sap-feeding insect vectors[15]. Over long distances, human activities are responsible for the dispersal of this pathogen through the trade of infected plant material. In the recent past, *X. fastidiosa* has been introduced in Taiwan via infected grapevine for planting[16]. Trade of ornamental coffee plants, from Central America to Italy, is the presumed means of introduction of the strain decimating Apulia's olive trees[17]. Several lines of evidence indicate that, *X. fastidiosa* strains responsible for the Pierce's disease of grapevine, that still represents a strong constraint for the Californian vineyards, were introduced from Central America via infected coffee plants[4]. Transport of infected *Prunus* was the likely vector of the introduction of strains from the subspecies *multiplex* from North America to Southern Brazil[18]. The United-States population would also have served as a reservoir for introductions to Taiwan and Europe[19].

To date, in Europe the three subspecies of *X. fastidiosa* were detected in four different countries i.e. in France, Italy, Portugal and Spain, with a large set of lineages: subsp. *fastidiosa* ST1, ST2, ST72 and ST76, subsp. *multiplex* ST6, ST7, ST81, ST87, ST88 and ST89, and subsp. *pauca* ST53, ST80[8,20–25]. All these records of different STs in distant areas are clear indications of various independent introductions of *X. fastidiosa* strains in Europe[21]. Many of these STs had previously been described in the Americas that hence represents their most probable area of origin and source of introduction in Europe[20,21,26]. These hypotheses were tested and confirmed by a first analysis of a limited number of strains from most EU outbreaks[22].

In France, a first suspicion of *X. fastidiosa* infection dates back from 1989, when a grapevine plant was found infected in the Languedoc region[27]. Then, in 2010 in the Saint-Emilion area, a genomic signature of the bacterium was found in the microbiota of a grapevine plant[28]. These two first observations did not lead to any *X. fastidiosa* establishment. *X. fastidiosa* presence in France has been declared officially in 2015, when surveillance led to its discovery in Corsica and Provence-Alpes-Côte d'Azur (PACA) regions[21]. It was subsequently detected in 2020 in Occitanie[25]. More than 99% of these foci were infected by the subsp. *multiplex* ST6 and ST7 lineages. While a large number of foci were detected all over Corsica (354 foci leading to a containment status in 2017) in the Corsican bush and in urban areas and on a large range of endemic or introduced plant species (39 hosts in 2018 and 49 in 2022), a limited number (166 foci) of mostly urban foci were reported in PACA on a range of 25 species, from which 15 plant species were not reported infected in Corsica. The pathways of invasion in these two regions remain to be elucidated.

MLST is one of the most popular technique to genotype pathogenic bacteria. It was widely used to resolve the epidemic spread of pathogens both in human, animals and plants[29–31]. Nevertheless, this method lacks resolution for epidemiology analyses within each ST as strains are undistinguishable[32,33] and MultiLocus Variable number of tandem repeat Analysis (MLVA) may then be preferred. MLVA is based on the analysis of rapidly evolving markers allowing to study recent events and discriminate individuals with more resolution than MLST[34]. MLVA allowed deciphering population genetic structure in the monomorphic bacteria *Xanthomonas citri* pv. *citri* and *Xanthomonas citri* pv. *viticola*[33,35] and to infer their invasion routes using Approximate Bayesian Computation (ABC)[35].

Assuming an exogenous origin of *X. fastidiosa* subsp. *multiplex* in France, our aim was to assess the genetic relatedness between ST6 and ST7 strains originating from Corsica and PACA, and decipher their most probable scenario of introduction, on the basis of the plant material gathered in the frame of the official French monitoring plan between 2015 and 2018. We mainly based our study on MLVA to take advantage of both the large sample collection and the capacity of these markers to monitor recent evolutionary events to reach the resolution required for subsequent analyses. Bayesian methodologies were used to infer the number of introductions and their most probable scenario of population evolution and spread. Simultaneously, genome data were used to date the divergence of French *X. fastidiosa* subsp. *multiplex* strains from their American relatives. Beyond the study of the introduction of *Xylella* in France, we present an interdisciplinary approach adaptable to the study of any new disease emergence.

## Results

**A low polymorphism within ST6 and ST7 genome sequences but enough temporal signal to date divergence time.** A set of 82 genome sequences of *X. fastidiosa* subsp. *multiplex* composed of all publicly available ones, as well as newly acquired sequences were analyzed in this study (Supplementary Data 1). They represented diverse geographical origins (Brazil $n = 1$, France $n = 52$, Italy $n = 3$, Spain $n = 12$, USA $n = 14$), spanned over 36 years of evolution (1983–2018), and were centered on the most frequent lineages detected in France, i.e., ST6 and ST7. Most genome sequences (85.4%) were Illumina dye sequences. Average genome length was 2,520,537 bp, with a mean N50 and L50 of 354,785 bp and 7.28, respectively. The length of their core genome alignment was 1,679,574 bp, in which 16,739 single nucleotide polymorphisms (SNPs) were detected. After removal

of regions with evidence of recombination 13,818 SNPs only originating from mutation events were kept.

Considering ST6 and ST7 strains genome clustering from all geographic origins using FastSTRUCTURE on non-recombinant SNPs, the patterns find was in partial discordance with traditional MLST grouping (Fig. S1). While all French and American ST6 strains clustered in a homogeneous cluster, as expected from MLST, Spanish ST6 strains were scattered within the ST7 cluster grouping all the French ST7 strains, the three American ST7 strains, and one of the Spanish ST81 strain. The two other Spanish ST81 strains were intermediate between French ST7 and the Spanish ST6 strains. American, Italian and Brazilian *X. fastidiosa* subsp. *multiplex* strains of other STs grouped into three other clusters (Fig. S1). Similar groupings were observed by plotting the core genome SNPs from mutation on a heatmap (Fig. S2). The low polymorphism observed in the core genome and the absence of sub-structuration within French ST6 or ST7 genome sequences were not compatible with any further population genetics analysis aiming at reconstructing the pathways of the emergence of *X. fastidiosa* at a regional scale in French territories based only on these data.

In contrast, the complete set of 82 genome sequences, isolated all over the world, was proved suitable for investigation of the timing of *X. fastidiosa* subsp. *multiplex* divergence (Supplementary Data 1). Analyzing the linear regression of sample age against root-to-tip distance (Fig. S3) and performing a date randomization test (DRT) with BEAST[36] (Fig. S4) revealed a sufficient temporal signal at the whole-tree scale. Therefore, the dataset could be used to coestimate evolutionary rates with ancestral divergence times with a tip-based calibration approach.

At the scale of the genome, the mean substitution rate for *X. fastidiosa* subsp. *multiplex* was inferred at $3.2165 \times 10^{-7}$ substitutions per site per year (95% Highest Posterior Density [HPD] $1.5468 \times 10^{-7}$–$5.2043 \times 10^{-7}$ substitutions per site per year). The standard deviation of the uncorrelated log-normal relaxed clock calculated by BEAST was 1.020, suggesting moderate variation in rates among branches. The divergence between French and American ST6 strains was estimated in 1987 (1974–1994 95% HPD) (Fig. 1). Then the divergence between these strains and the Spanish ST81 strains was estimated in 1425 (766–1914 95% HPD). The one between ST7 French and American strains was estimated in 1971 (1924–1994 95% HPD). This time to the most recent common ancestor (TMRCA) was due to the USA RAAR6Butte strain, as the TMRCA with the two other ST7 USA strains (M12 and Griffin-1) is older (=1510; 991–1911 95% HPD). The split between Spanish ST6 strains and ST7 strains was estimated in 1027 (120–1743 95% HPD). The divergence between the group of ST7 and Spanish ST6 strains and the group of ST81 and ST6 French and American strains was estimated in 755 (−397 to 1540 95% HPD).

**The development of a MLVA efficient on DNA extracted from plant samples revealed that French *X. fastidiosa* split into four groups of strains**. To elucidate the scenario that led to the establishment of *X. fastidiosa* subsp. *multiplex* in Corsica and PACA, variable number of tandem repeats (VNTRs) were used to complement the low information gained from SNP calling. An in silico analysis of the *X. fastidiosa* subsp. *multiplex* strain M12 genome sequence was performed to search for new VNTRs, in order to complete the set of available ones[37,38]. A set of 13 VNTRs was selected and while initially developed to discriminate strains of the subspecies *multiplex*, it proved to be valid for use on all other *X. fastidiosa* subspecies (Table 1, Supplementary Data 2, Fig. S5). The developed VNTR-13 scheme was then optimized for direct use on DNA extracted from plant samples, due to the

difficulty of isolating the strains and to make use of the large amount of infected plant samples available in France. The complete setup of the VNTR-13 scheme including its validation is detailed as supplementary results.

In France, from 2015 to 2018, among the plant samples tested in the framework of the French surveillance and officially declared to be *X. fastidiosa* infected, 917 samples had a Cq<32 (the amplification limit Cq of the MLVA), which corresponded to $\sim5 \times 10^5$ *X. fastidiosa* cells per gram of plant. Depending on the availability of frozen samples and in order to avoid the analysis of several samples from the same foci, a selection of samples was made to maximize the number of foci and plant species analyzed and resulted in a set of 534 samples for the MLVA. A total of 396 samples were successfully genotyped for all 13 loci (184 ST6 and 212 ST7; i.e. 43.18% of the French *X. fastidiosa* samples available and 74.16% of the tested samples) (Table 2, Supplementary Data 3). The loci were all highly polymorphic across the French dataset with a number of alleles ranging from 7 to 21 and a number of TRs ranging from 1 to 30 (Table 3). Simpson's diversity index ranged from 0.52 to 0.88 and allelic richness from 3.82 to 15.73. For the VNTR loci ASSR-19, XFSSR-40 and XFSSR-58, all possible allele sizes of the allelic range were observed within the collection. For the VNTR loci ASSR-9, ASSR-11, ASSR-12, ASSR-16, COSS-1, GSSR-7, OSSR-16, OSSR-19, XFSSR-37 the observed diversity of TR sizes did not cover the entire allelic range as one to three TR sizes were not observed, indicating either missing infected samples in the evolution path or large mutation steps (Supplementary Data 3). MLVA accurately resolved the different haplotypes from the French outbreaks as more than 95% of the haplotypes were detected with 12 markers (Fig. S6). The discriminatory power of the MLVA was 0.9969, showing a very high level of discrimination.

For a minority of 13 French *X. fastidiosa* infected plant samples, several peaks were observed on the electrophoregrams on 3 up to 12 amplified loci, and this was confirmed in at least two independent analyses. As some alleles were, for now, specific of ST6 or ST7 (e.g., for ASSR-16:29 TR = ST6, 24 TR = ST7), these results revealed the presence of co-infections by several strains in these plants and for some of them potentially by several STs.

MLVA allowed observing a large haplotype diversity within French ST6 and ST7 *X. fastidiosa* as 320 haplotypes were delineated, among the 396 samples (Supplementary Data 3). The 184 ST6 samples were genotyped in 148 haplotypes, while the 212 ST7 samples were genotyped in 172 other haplotypes, and no VNTR profile was shared between these two STs (Supplementary Data 3 and 4, Fig. S7). The distribution of allele frequencies for each of the 13 VNTR loci did not indicate differences between samples isolated in Corsica or PACA or their host plant. ST6 samples were grouped into four clonal complexes (i.e., networks grouping haplotypes differing from their closest neighbor at a single VNTR locus). The founder haplotype (#309) of the main clonal complex grouped 11 samples from three plant species (*P. myrtifolia*, *Lavandula* x *allardii*, *Calicotome villosa*), and was linked, in this clonal complex, to 95 samples that were all sampled in Corsica (Fig. 2). ST7 samples grouped into 15 clonal complexes. The founder haplotype (#138) of the main clonal complex comprised 15 samples of three plant species (*P. myrtifolia*, *Genista* x *spachiana*, *Helichrysum italicum*) and was linked to 88 samples, of which 86 were isolated in Corsica and only two in PACA. The other 17 smaller clonal complexes grouped from two to eight samples that were sampled in a same region, with the exception of one clonal complex that grouped haplotype #163 sampled in PACA and haplotype #165 sampled in Corsica. The remaining 167 samples represent singletons, differing by at least two loci (=80 ST6 haplotypes and 83 ST7

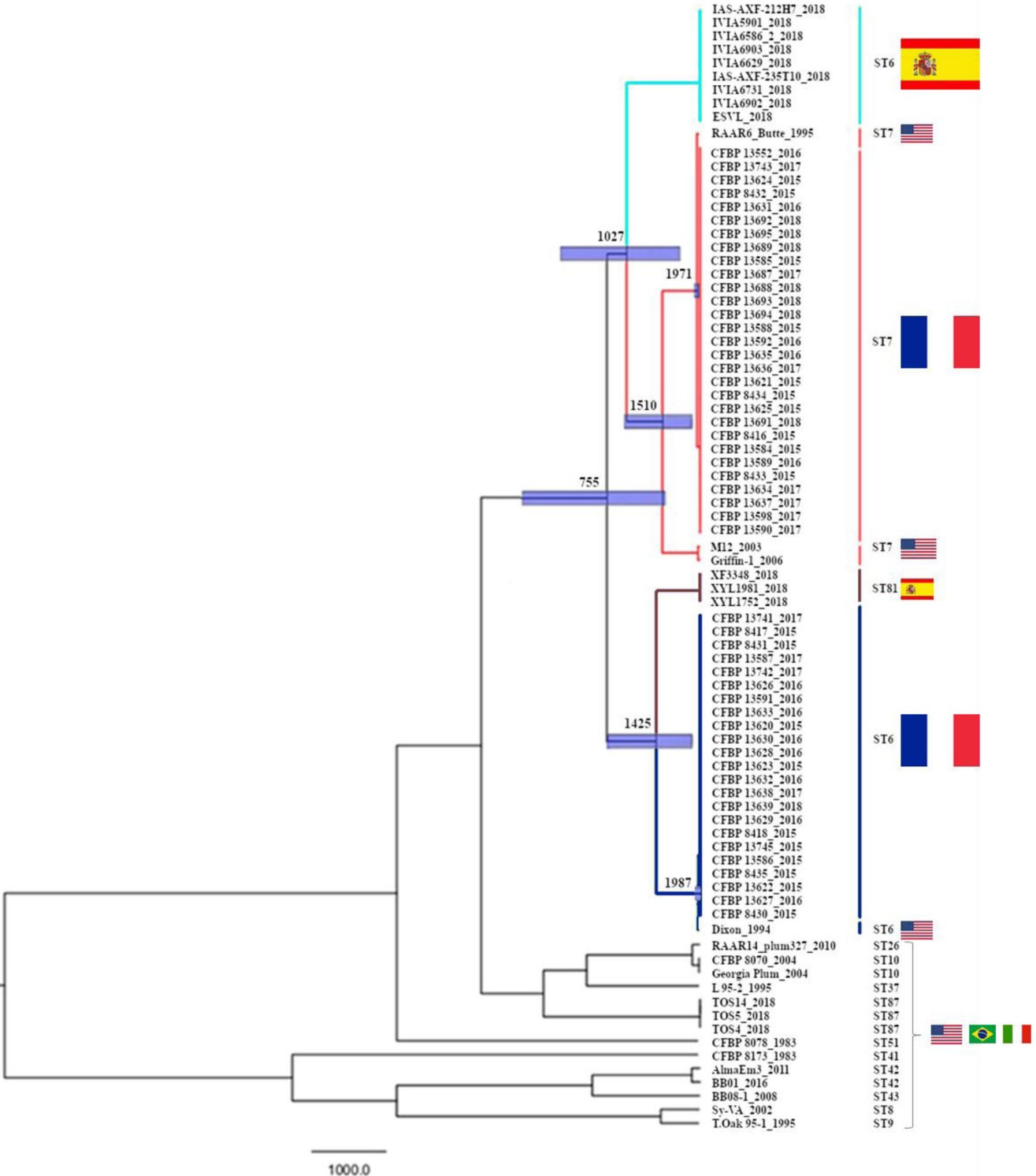

**Fig. 1 Phylogenetic tip-dated tree showing the estimated divergence date between *X. fastidiosa* subsp. *multiplex* strains (*n* = 82).** Genealogy was inferred by Maximum-likelihood phylogenetic inference using BEAST 2.6.1. and a GTR model, based on SNPs variations among the 82 genome sequences. For details on the data, refer to Supplementary Data 1. The tree main sequence types are highlighted in blue (ST6), red (ST7) and brown (ST81) and flags refer to the country of isolation of the strains. Node bars refer to 95% HPD.

haplotypes) or by three and more loci (=36 ST6 haplotypes and 32 ST7 haplotypes) (Supplementary Data 3).

Due to the nature of the data that mostly came from official monitoring programs, which aim is the eradication of any infected plant and not population genetics studies, it was impossible to analyze the impact of the host and year of sampling on the minimum spanning tree (MST) structure. Nevertheless, the presence of infected samples from 2015 in distal parts of evolutionary branches and of founder haplotypes sampled in 2017 (Fig. 2) suggests that sampling was carried out in an already diversified population. Regarding host plants, a large majority (69.95%) of our dataset was made of myrtle-leaf milkwort plants (*P. myrtifolia*), while Spanish broom (*Spartium junceum*) accounted for 7.07% of the samples and then all remaining 23

**Table 1 Nomenclature, location, function and genetic diversity of the 13 TR loci.**

| Locus | Primers | Repeat motif | On the M12 genome sequences | | | | On the 10 genome sequences of X. fastidiosa subsp. multiplex | | | Reference |
| | | | Number of motif repetition | Location | Coordinates (start - end) | Multiplexing annealing temperature (°C) | Number of alleles | Range of repetition number | Simpson's diversity index | |
|---|---|---|---|---|---|---|---|---|---|---|
| ASSR-11 | ATTO550 - AGAGGCAACGCAGGAACAG GTGAGTTATATCGGTGCAGCAG | ACGCATC | 14 | Hypothetical protein | 2,074,633-2,074,920 | 53 | 6 | 3-14 | 0.18 | Lin et al., 2005 |
| ASSR-16 | ATTO565 - TTAATCAACAACGCTTATCC TCGCAGTAGCCAGTATAC | GCTCCGGTTCTA | 19 | 1,4-beta-cellobiosidase CDS | 658,561-658,173 | 53 | 8 | 8-29 | 0.16 | Lin et al., 2005 |
| COSS-1 | 6-FAM - GAAACAAGATGGCGGTTGC CATTTAAAGCGGGCGCATA | ATTGCTG | 2 | Non coding | 18,250-18,484 | 53 | 7 | 1-11 | 0.16 | Francisco et al., 2017 |
| OSSR-19 | HEX - GCTGTGAACTTCCATCAATCC GCAAGTAGGGGTAAATGTGAC | CAGGATCA | 7 | adenylosuccinate lyase CDS | 967,661-967,412 | 53 | 6 | 3-11 | 0.2 | Lin et al., 2005 |
| GSSR-7 | ATTO550 - ATCATGTCGTGTCGTTTC CAATAAAGCACCGAATTAGC | AGCAAC | 11 | Non coding | 19,289-19,595 | 55 | 8 | 9-25 | 0.14 | Lin et al., 2005 |
| OSSR-16 | ATTO565 - GCAAATAGCATGTACGAC GTGTTGTGTATGTGTTGG | CTGCTA | 9 | Non coding | 593,243-593,502 | 55 | 6 | 7-17 | 0.24 | Lin et al., 2005 |
| XFSSR-37 | HEX - CACCTGCAACGAACACCAAT TTCAGTTGGTGTTGCAGGTC | CTGATGTG | 8 | hydroxyacylglutathione | 1,421,433-1,421,644 | 55 | 4 | 3-12 | 0.3 | This study |
| XFSSR-58 | 6-FAM - CCCTAGCAAAACAATACTGCCA GGTATGTGCTGTGTGGTTGG | AAGGGGC | 7 | hydrolase CDS Non coding | 1,997,325-1,997,556 | 55 | 6 | 3-12 | 0.22 | This study |
| ASSR-9 | 6-FAM - GGTTGTCGGGCTCATTCC TTGTCACAGCATCACTATTCTC | CAAGTAC | 9 | Non coding | 949,863-950,106 | 57 | 6 | 3-15 | 0.22 | Lin et al., 2005 |
| ASSR-12 | ATTO565 - TGCTCATTGTGGCGAAGG CGCAACGTGCATTCATCG | GATTCAG | 9 | Hypothetical protein | 2,116,062-2,115,798 | 57 | 7 | 4-14 | 0.16 | Lin et al., 2005 |
| ASSR-19 | ATTO550 - CGCCGACTGTCTATGTGAC TTCCTTAGCAATGGCAATGTTG | ACAACG | 3 | Hypothetical protein | 2,075,410-2,075,699 | 57 | 7 | 2-12 | 0.16 | Lin et al., 2005 |
| GSSR-4 | HEX - GCGTTACTGGCGACAAAC GCTCGTTCCTGACCTGTG | ATCC | 4 | Hypothetical protein | 1,702,352-1,702,608 | 57 | 3 | 3-5 | 0.38 | Lin et al., 2005 |
| XFSSR-40 | 6-FAM - ACACACTCACACTGTCCGAT GGGTTAGGAGTTGGTGTCGA | ACAGCAAT | 5 | Non coding | 1,513,631-1,513,860 | 60 | 6 | 3-11 | 0.2 | This study |

**Table 2 Characteristics of the 396 French strains and plant samples used in this study.**

| Summary per host of isolation | Corsica | PACA | Total |
|---|---|---|---|
| *Acacia* sp. | / | 1 | 1 |
| *Artemisia arborescens* | 1 | / | 1 |
| *Calicotome villosa* | 15 | / | 15 |
| *Cistus* spp. | 5 | / | 5 |
| *Convolvulus cneorum* | / | 3 | 3 |
| *Coronilla* spp. | 1 | 4 | 5 |
| *Cytisus villosus* | 2 | / | 2 |
| *Euryops chrysanthemoides* | / | 8 | 8 |
| *Genista* spp. | 6 | / | 6 |
| *Grevillea juniperina* | / | 1 | 1 |
| *Hebe* sp. | 1 | / | 1 |
| *Helichrysum* spp. | 12 | 3 | 15 |
| *Lavandula* spp. | 7 | 1 | 8 |
| *Lonicera japonica* | / | 1 | 1 |
| *Medicago* sativa | / | 1 | 1 |
| *Metrosideros excelsa* | 1 | / | 1 |
| *Pelargonium* spp. | 5 | / | 5 |
| *Phagnalon saxatile* | 1 | / | 1 |
| *Polygala myrtifolia* | 205 | 72 | 277 |
| *Prunus* spp. | 3 | 3 | 6 |
| *Quercus suber* | 1 | / | 1 |
| *Rosa canina* | 1 | / | 1 |
| *Spartium junceum* | 16 | 12 | 28 |
| *Veronica* spp. | / | 2 | 2 |
| *Westringia fruticosa* | / | 1 | 1 |
| Summary per year of isolation | | | |
| 2015 | 166 | 14 | 180 |
| 2016 | 75 | 4 | 79 |
| 2017 | 34 | 39 | 73 |
| 2018 | 8 | 56 | 64 |
| Summary per sequence type | | | |
| ST6 | 154 | 30 | 184 |
| ST7 | 129 | 83 | 212 |

**Table 3 Genetic diversity based on 13 VNTRs in the 396 French samples.**

| Locus | Nb of alleles | Nb of repetition min–max | Simpson's diversity index (1-D) | Allelic richness |
|---|---|---|---|---|
| ASSR-11 | 11 | 5–16 | 0.69 | 7.79 |
| ASSR-16 | 7 | 22–30 | 0.52 | 8.41 |
| COSS-1 | 11 | 1–12 | 0.76 | 5.25 |
| OSSR-19 | 11 | 3–14 | 0.64 | 4.28 |
| GSSR-7 | 19 | 10–29 | 0.88 | 13.46 |
| OSSR-16 | 21 | 6–29 | 0.75 | 9.45 |
| XFSSR-37 | 8 | 1–10 | 0.56 | 6.88 |
| XFSSR-58 | 8 | 4–11 | 0.54 | 6.71 |
| ASSR-9 | 13 | 5–18 | 0.82 | 4.98 |
| ASSR-12 | 7 | 7–14 | 0.62 | 7.86 |
| ASSR-19 | 8 | 7–14 | 0.74 | 3.82 |
| GSSR-4 | 7 | 8–21 | 0.52 | 15.73 |
| XFSSR-40 | 8 | 5–12 | 0.73 | 6.1 |

subgroups when the corresponding populations were differentiated based on $R_{ST}$ and $F_{ST}$ pairwise comparisons, with a *p*-value<0.05 (Supplementary Data 3 and 5). Moreover, analysis of molecular variance evaluated that the majority of the genetic variance occurred within the subgroups (ST6: 63.99%, ST7: 78.12%) (Supplementary Data 6). As a result, three groups were defined for each ST (i.e. one American and two French, Supplementary Data 3, Fig. 4). For ST6, the DAPC cluster 1 included all samples of the main clonal complex plus a few singletons. This group of samples was mainly isolated from Corsica (135 samples) and a few from PACA (two samples) and was named ST6_C1P1. The DAPC cluster 2 was composed of all the other singletons and two of the small clonal complexes grouping two haplotypes. It was separated into two subgroups: one named ST6_C2, grouping the 19 samples originating from Corsica and another one named ST6_P2 grouping the 28 samples originating from PACA. For ST7, the DAPC cluster 3 included all samples of the main clonal complex plus a few singletons. This group of samples were mainly isolated from Corsica (110 samples) and a few from PACA (eight samples) and was named ST7_C1P1. The DAPC cluster 4 was composed of all the other singletons and 13 of the small clonal complexes of two haplotypes. The DAPC cluster 4 was separated into two subgroups, one, named ST7_C2, composed of the 19 samples originating from Corsica, and another one, named ST7_P2, composed of the 75 samples originating from PACA.

The number of scenarios with three French populations, one outgroup population and possible non-sampled (ghost) populations is huge, making it impossible to perform a single analysis to answer our question. To cope with this complexity, we adopted a two-step approach composed of (i) a bottom-up step in which populations from a same ST were analyzed two by two in three different analyses, which aimed at deciphering histories between pairs of French populations (Fig. S9A), and (ii) a top-down step confronting three-population scenarios not excluded by the bottom-up step (Fig. S9B).

On the three **French ST6 populations** (C1P1, C2, and P2), after bottom-up approach, no scenario (topology × combination of populations) achieved sufficiently high posterior probability and low prior error rate to be considered as the best scenario to reconstruct the evolutionary history of the ST6 French populations in regards to their USA counterparts. However, some scenarios had so low posterior probabilities (*p*-value < 0.05) that they could be ruled out (Supplementary Data 7). Specifically, scenarios 2 and 6 testing the possibility of only one French

plant species each accounted for less than 3.78% of the samples, which was highly imbalanced and did not allow the analysis of MST structuration relative to host species. Moreover, 4.68% of the haplotypes were identified in more than one plant species, indicating that haplotypes did not face intrinsic dispersal barriers.

In order to retrace the routes of dissemination, populations that can be analyzed by Bayesian methods were sought. On the basis of genetic clustering analyses (DAPC and STRUCTURE) and geographical information, we validated the clustering of all 396 samples into four clusters that were, as expected, also supported (*p*-value < 0.05) by $R_{ST}$ and $F_{ST}$ pairwise comparisons (Fig. 3, Fig. S8, Supplementary Data 5, See supplementary results for complete description of the clusters).

**Bridgehead introductions of *X. fastidiosa* subsp. *multiplex* ST6 and ST7 in Corsica from PACA.** We inferred the routes of dissemination of *X. fastidiosa* subsp. *multiplex* in France using Approximate Bayesian Computation, beginning with the definition of a set of evolutionary scenarios that may explain the observed data. Firstly, in order to keep tested scenarios as simple as possible, we analyzed ST6 and ST7 French samples separately, because there was no known element indicating that these different strains were introduced simultaneously into France and tip-dating provided different dates of divergence for French ST6 and ST7 strains from their American relatives. Secondly, we had to define groups of strains likely to have shared common history (in sexual species, these would be populations, but in bacteria this step is not trivial). In each cluster previously defined by DAPC at $k = 4$, samples originating from distinct regions were split as

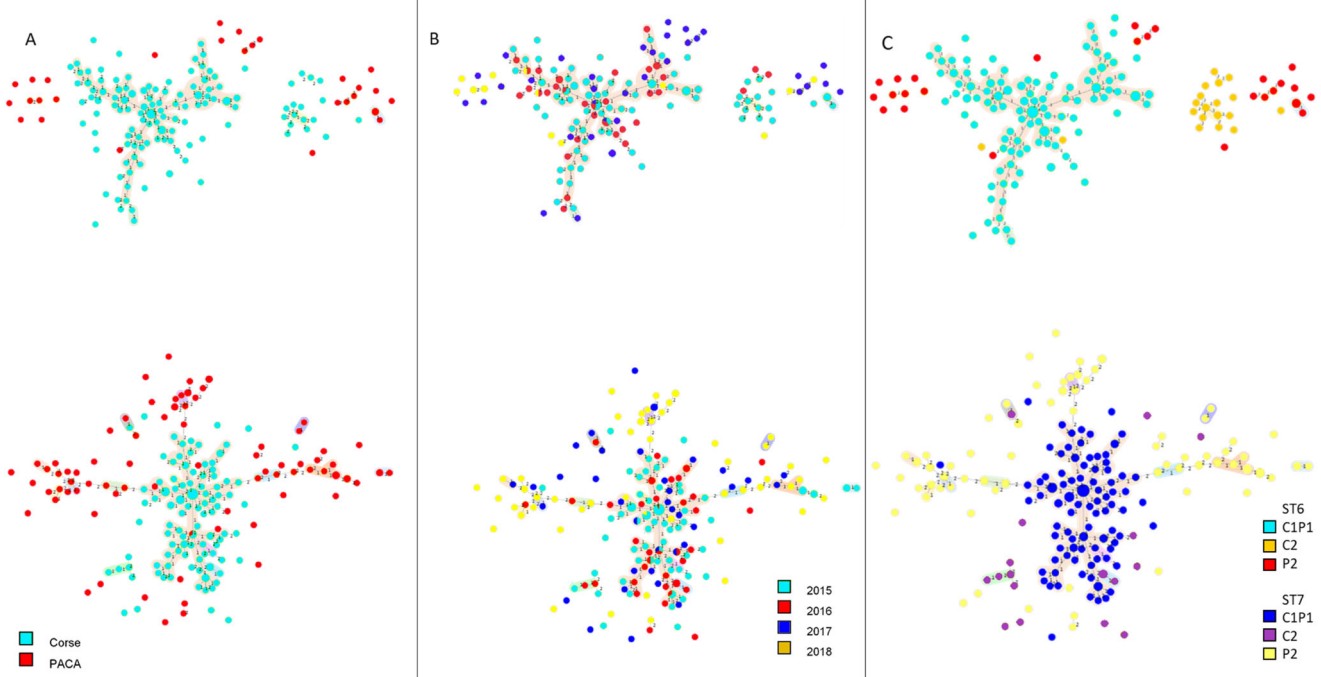

**Fig. 2 Minimum spanning trees of the 396 French *X. fastidiosa* subsp. *multiplex* infected samples typed using the VNTR-13 scheme.** Dot diameter represents the number of strains per haplotype. Link number refer to the number of TR loci polymorphic and distinguishing two haplotypes. Dot colors refer to **A** sampling region; **B** year of sampling; **C** ABC grouping. Colored area around haplotypes the clonal complexes.

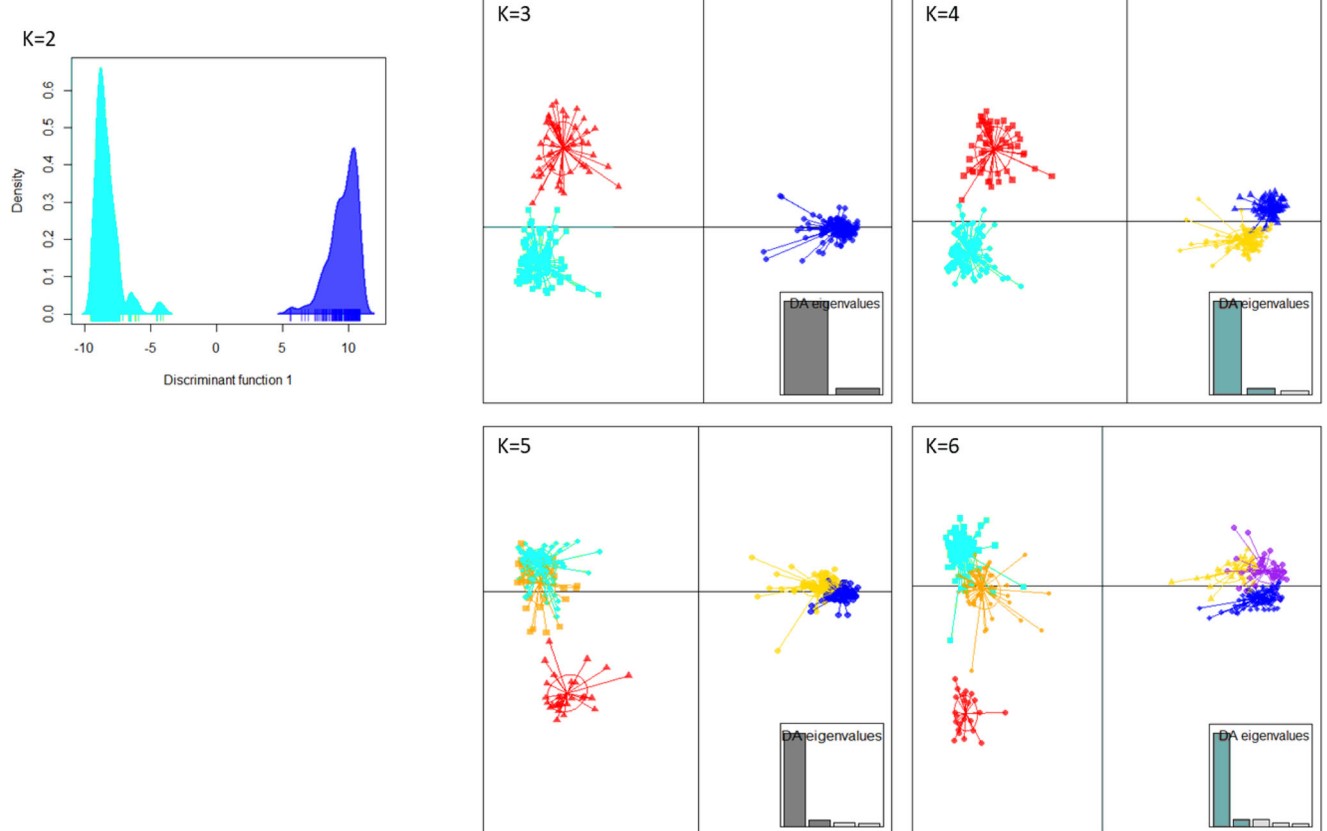

**Fig. 3 Scatterplot representing the Discriminant Analysis of Principal Components clusters for the 396 French *X. fastidiosa* infected samples for *k* = 2 to 6 inferred by use of the VNTR-13 scheme.** The eigenvalues showed that the genetic structure was captured by the first two principal components retained onto axis 1 (horizontal axis) and axis 2 (vertical axis). The dots represent the individuals, and the clusters are shown as inertia ellipses. Clusters turquoise, red and orange grouped ST6 samples and clusters blue, yellow, and purple grouped ST7 samples.

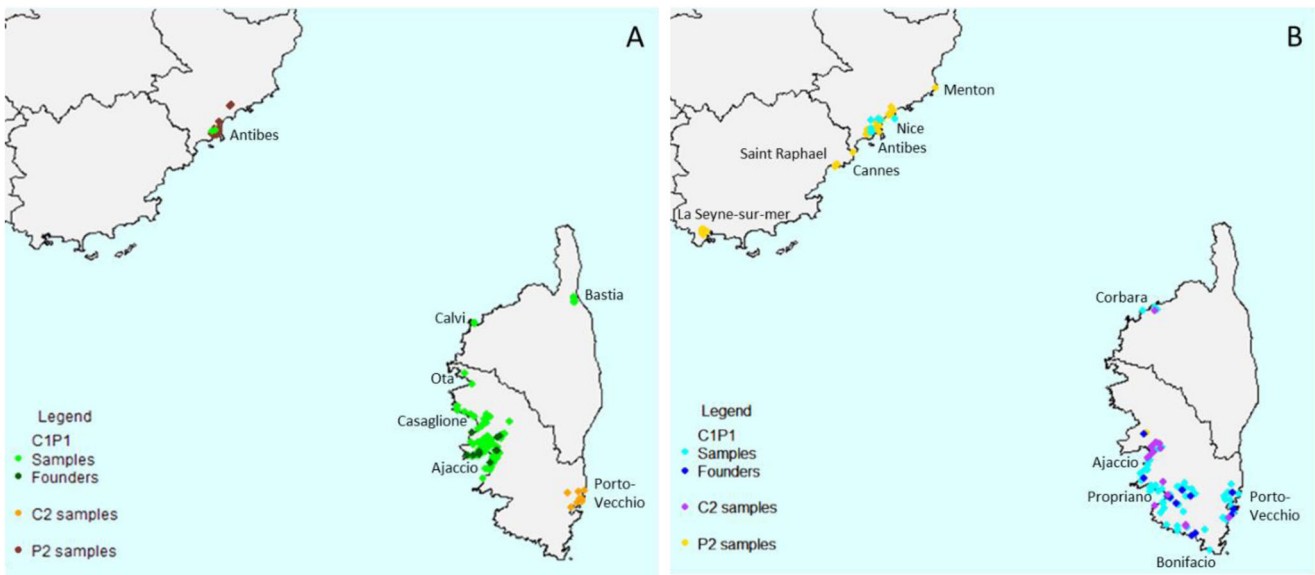

**Fig. 4 Distribution of the French *X. fastidiosa* infected samples used in this study. A** ST6 plant samples and strains, **B** ST7 plant samples and strains.

population were ruled out for all combinations of populations, confirming that the French ST6 samples did group into three populations (Fig. S9A).

Scenarios not excluded during this first step were used in the top-down approach to elaborate scenarios of the evolutionary history of the three French ST6 populations (Fig. S9B). Considering all combinations of populations, 30 scenarios were confronted (Fig. S10). Scenario II.7 was selected in the abcrf analysis as the most probable scenario with 9.9% of the votes and the highest posterior probability (0.26–0.37) (Supplementary Data 7). In this scenario, ST6 strains would have been first introduced in PACA (ST6_P2 population). Then population ST6_C1P1 would have diverged from this initial focus, and would thus represent the first established population in Corsica. A second independent population (ST6_C2) would have diverged later on from the first established ST6_P2 population in PACA, and established in Corsica (Fig. S10). However, it should be tempered by the fact that this selected scenario also presented a high prior error rate (0.72) (Supplementary Data 7).

The following most probable scenarios received a low number of votes. The second and third best scenario were III.I.16 and II.9 with respectively 5.8% and 5.7% of votes (Supplementary Data 7). These two scenarios had a close topology to scenario II.7, as both considered an introduction in PACA (ST6_P2), from which the population ST6_C2 from Corsica would have diverged. Moreover, on the 30 scenarios confronted, the seven scenarios testing the hypothesis that the Corsican population ST6_C2 would have diverged from the ST6_P2 population from PACA totalized 35.9% of the votes, giving more strength to this event.

Similarly, on the three French ST7 populations (C1P1, C2, and P2), the bottom-up approach did not allow to identified the best scenario, but allowed ruling out scenarios 2 and 6, thereby confirming that the French ST7 samples did group into three populations (Fig. S9A, Supplementary Data 7). In addition, scenarios 1 and 9 that tested independent and successive introductions of each French populations from the American ancestor were also excluded. It is however interesting to note that similar scenarios (5 and 10) including an unsampled population between the American ancestor and the French ones were not ruled out. In contrast, scenarios 11 and 12 that were slight

variations of previous scenarios 1 and 9 including unsampled populations between the American ancestor and each French populations were ruled out. Altogether, six scenarios (no. 3, 4, 5, 7, 8 and 10, Fig. S9A) were kept for further consideration after this bottom-up approach.

Moving to the top-down approach, based on the results of the bottom-up approach scenarios that tested independent introductions (class I in Fig. S9B) and those testing two independent introductions, one of which was responsible for a bridgehead invasion of the third population (class III.I and III.II in Fig. S9B), were consequently not considered. This left us with two closely related topologies to be tested (Fig. S9B and Fig. S11), i.e., scenarios testing the introduction in France of one population subsequently responsible for two independent bridgehead invasions, (class II in Fig S10B), and scenarios considering that the first introduction lead to two successive bridgehead invasions (class IV in Fig S10B). Considering all combinations of populations, 12 scenarios were confronted. Scenario II.7 was selected in the abcrf analysis as the most likely scenario with 17.5% of the votes and with the highest posterior probability (0.41–0.50) (Supplementary Data 7). In this scenario, ST7 strains would have been first introduced in PACA (ST7_P2 population). Then population ST7_C1P1 would have diverged from this initial focus, and would thus represent the first established population in Corsica. A second independent population (ST7_C2) would have diverged later on from the first established ST7_P2 population in PACA, and established in Corsica (Fig. S11). However, it should be tempered by the fact that this well supported scenario also presented a high prior error rate (0.70–0.71) (Supplementary Data 7).

Then scenarios having the highest number of votes were scenario IV.27 and scenario IV.25, to which respectively 12.4% and 10.9% of the votes were attributed (Supplementary Data 7). Both had a close topology to the best scenario, as they considered that a first population would have been introduced in PACA (ST7_P2), from which populations ST7_C1P1 and ST7_C2 would have diverged but from two successive bridgehead invasions (ST7_C1P1 then ST7_C2 or ST7_C2 then ST7_C1P1) instead of two independent divergence events. Moreover, among the 12 scenarios confronted, the four scenarios testing the hypothesis that the population from PACA (ST7_P2) was the first one

introduced in France and from which the other two population diverged totalized 53.24% of the votes, giving more strength to this event.

## Discussion

While natural spreading capacities of *X. fastidiosa* depend on the activity of the sap-sucking insects that vector it from plant to plants, the detection of strains outside their areas of origin since the end of the 19th century illustrates the effectiveness of its human-assisted transmission, as elegantly demonstrated for *X. fastidiosa* strains introduced in the Balearic islands[39]. The introduction of infected but asymptomatic plants for planting of species not cultivated in Europe, such as coffee plants, or of various ornamental species is now considered one, if not the main pathway for introduction of *X. fastidiosa*[11]. The introduction of the subspecies *fastidiosa* in California would have been the first example[4,40] and the introduction of *X. fastidiosa* subsp. *pauca* strain ST53 into Apulia, Italy, the most recent one described[22]. It is within this framework that we have sought to reconstruct the invasive scenario of *X. fastidiosa* subsp. *multiplex* ST6 and ST7 in France.

First, the molecular evolutionary rates of *X. fastidiosa* subsp. *multiplex* was evaluated using tip-calibrating approach at $3.2165 \times 10^{-7}$ substitutions per site per year ($1.5468 \times 10^{-7}$–$5.2043 \times 10^{-7}$ 95% HPD). This estimation is congruent with those previously estimated for the subspecies *pauca* ($7.62 \times 10^{-7}$ substitutions per site per year, 95 confidence interval (CI): $1.027 \times 10^{-7}$ to $1.454 \times 10^{-6}$) and for the subspecies *fastidiosa* ($6.38 \times 10^{-7}$ substitutions per site per year, 95 CI: $3.9277 \times 10^{-7}$ to $9.0912 \times 10^{-7}$ and $7.71 \times 10^{-7}$ substitutions per site per year, 95 CI: $1.20 \times 10^{-7}$ to $1.69 \times 10^{-6}$ substitutions per site per year)[39,41,42]. Moreover, the mean substitution rate estimated in our study is consistent with those of several human and animal bacterial pathogens estimated within a similar time period[43]. This rate served to estimate the divergence time of strains from the main lineages present in France (ST6 and ST7) from their American and Spanish relatives. Divergence date between French and American ST6 strains was estimated in 1987 (1974–1994 95% HPD), while divergence date between French and American ST7 strains was estimated earlier in 1971 (1924– 1994 95% HPD). Moreover, very few SNPs were detected between French strains sharing the same ST, confirming recent introductions, even if divergence time is not synonymous to introduction time, but here represent the lower bounds of introduction. The period of divergence of ST6 and ST7 French strains from US counterparts corresponds to the 70's-80's, a period during which alien plants were massively introduced in Corsica[44]. To our knowledge, no such data about alien species trade, in particular from the only other known place of occurrence of *X. fastidiosa* subsp. *multiplex* ST6 and ST7, the USA, could be recovered to document potential introductions in PACA. The period of time for *X. fastidiosa* introduction into France inferred from analysis of genomic data is highly consistent with a previous evaluation dating it in 1985 (1978–1993 95% HPD) using a mechanistic-statistical approach based on spatio-temporal modeling of the French surveillance plan data and the assumption of a hidden compartment limiting the exhaustiveness of the sampling[45]. In contrast, our results are not consistent with estimates from the study of Moralejo et al.[39], which dated the introduction in Corsica of *X. fastidiosa* subsp. *multiplex* in 2000. In this study only three Corsican strains (two ST6 and one ST7) all isolated in 2015 were simultaneously analyzed[39]. In our study, the dating of the French strains, make them the earlier ones that would have occurred in Europe, before the introduction of ST1 in 1993 and of ST81 in 1995 in the Balearic Islands, of ST6 in 2005 in mainland Spain and of ST53 in

Apulia[39,41]. We also confirmed the polyphyly of the French and Spanish ST6 strains previously observed[22]. These two groups of strains differed by 899 to 912 SNPs and their divergence date traced back to the divergence between ST6/ST81 and ST7 estimated in 755 (−397-1540 95% HPD). At the time of this study no strain or genome of Spanish ST7 strains was available, which impeded their inclusion in our study. Regarding the SNPs detected on the core genome of our dataset (1,679,574 bp containing 16,739 SNPs), data are in line with those previously obtained by Landa et al.[22], in which the core genome of 21 *X. fastidiosa* subsp. *multiplex* strains was estimated at 1,569,508 bp, containing 5630 core SNPs. But these data are surprisingly in contrast with those of Vanhove et al.[41] in which the core genome of 23 strains of *X. fastidiosa* subsp. *multiplex* was estimated at 736,868 bp containing 37,485 SNPs. Differences may rely on the use of different methodologies as Vanhove et al.[41] study analyzed mapped raw reads data on reference and both Landa's et al.[22] and our study analyzed assembled genomes which is more precise.

To trace back the most probable invasive scenario for ST6 and ST7 strains in France, we developed a MLVA typing scheme of 13 VNTRs optimized to be directly used on genomic DNA extracted from fresh or frozen infected plant material. A similar approach was also developed for Italian samples infected with subspecies *pauca* ST53[46]. Our study further confirms the interest of such direct genotyping techniques adapted for analyses at small evolutionary scales. The scheme composed of discriminant newly identified and existing VNTR loci[37,38], was able to type all *X. fastidiosa* subspecies, but was specifically developed and optimized to study subspecies *multiplex* diversity.

MLVA highlighted substantial diversity within each ST as three subgroups of samples were defined that partially fitted with the geographical location of samples. These groups however clustered samples from various plant species. On the one hand such genetic structures are highly indicative of a long distance spread of the pathogen by human activities for the groups mixing geographical origins of strains, i.e., lacking spatial structuration, and human activities are often associated with fortuitous introductions of pathogens into yet free areas[47]. On the other hand, the two other groups of each ST that are highly spatially structured are indicative of a more local dissemination that can result from initial human activities but also, as it overcomes host barrier, from the subsequent activity of insect vectors, the only natural means of dispersion of *X. fastidiosa*. In Corsica, *Philaenus spumarius* was identified as a potential vector of *X. fastidiosa* ST6 and ST7 strains and is widely spread in the island[48].

Population genetic analyses of the ST6 samples highlighted a first introduction of *X. fastidiosa* in PACA (ST6_P2) from America. It would have then spread within this region, where it was found on ornamental plants in urban contexts (limited area around cities Antibes and Nice). Then, a second population (ST6_C1P1) diverged from this bridgehead initial population and also established in PACA. Individuals were also introduced and established in Western Corsica, where the founders of this group were mainly found around the harbor area of the city of Ajaccio. Finally, the last Corsican population (ST6_C2) would have also diverged from the bridgehead population initially established in PACA (ST6_P2) and spread around Zonza in the Eastern part of the island. For now, ABC analyses did not fully validate all this scenario, but the hypothesis of a bridgehead scenario, in which the Corsican population (ST6_C2) would have diverged from the PACA population (ST6_P2) is the most likely to have happened.

The same scenario was found as the most probable one for ST7, the differences rely mostly in places of occurrence, with co-location of ST6 and ST7 infected samples in some places. The first introduction in Corsica (ST7_C1P1) was widely spread in Southern Corsica, while the second population (ST7_C2) spread

essentially around Ajaccio, but some foci were found in the Northern and Western parts of Corsica. This larger distribution over the Corsican territory is congruent with an older introduction (1971), estimated 16 years earlier than for ST6 (1987). Here again, ABC analysis did not fully validate the entire scenario, but the hypothesis according to which the population from PACA (ST7_P2) was the first established in France, from which the two other diverged from a bridgehead is the most likely to have happened. Bridgehead effect is a concept first developed for eukaryote invasive species[49] and recently applied to phytopathogenic bacteria[35]. This concept refers to widespread secondary invasions stemming from a particularly successful invasive population, which serves as a source of colonists for potentially remote new territories[50].

Here, we suspect that trade of infected myrtle leaf milkwort represents all initial migration events that served later on as a reservoir for local dissemination by plant movement or insect vectors, which would explain the identification of same haplotypes in plants of different species. This plant species represents 69.95% of the samples detected infected over the reference period, from which founder haplotypes of all clonal complexes were detected. *P. myrtifolia* is a bush native from South Africa, genetically bred in Florida in the 1980's, before being sold in Europe[51,52]. Moreover, the first identification of ST6 and ST7 of *X. fastidiosa* subsp. *multiplex* were made in the Eastern and South-Eastern USA[7]. In Europe, the three largest countries that produce *P. myrtifolia* cuttings are Italy, Spain and then Portugal[52]. Plants from southern Europe are sold to PACA, which serves as importer for Corsica, in which the trade of plants is unilateral, i.e. only entering[52]. In fact, when the *X. fastidiosa* detections arose in 2015 in Corsica, the origins of the first infected *P. myrtifolia* plants were Italy (8 plants), PACA (2 plants) and Spain (1 plant)[52]. Furthermore, the dates of divergence of the French ST6 and ST7 strains from the last common ancestor with their American counterparts, in 1987 and 1971 respectively, correspond to a period of massive introduction of exotic plants, particularly in Corsica[44]. Unfortunately, no such data were found for the PACA area plant trade. All these elements are in line with the best scenario that we obtained and could validate the myrtle leaf milkwort as the initial vector of *X. fastidiosa* subsp. *multiplex* in France. The climate all over France is supposed to be favorable for the spread and survival of *X. fastidiosa* subsp. *multiplex*[53]. A spread to other areas in France or the discovery of *X. fastidiosa* in other places, such as recently reported in Occitanie[24,25], should be anticipated. As the probable presence of *X. fastidiosa* in Corsica and PACA for half a century has been overlooked and it was identified through reinforced surveillance following its detection in Apulia. Indeed, in Corsica, sampling was quite exhaustive, as most infected samples grouped in clonal complexes that are biologically meaningful groups of single locus variants epidemiologically related. In contrast, most PACA infected samples grouped in several small clonal complexes and singletons that could not be linked as several evolutionary steps were lacking. Further sampling in the south of France would allow refining the scenario we presented here. Moreover, surveillance must be maintained or reinforced, depending of the regions, to avoid further introductions and the development of epidemics in this volatile context of climate change that could make conditions more favorable to *X. fastidiosa*.

## Methods

In order to trace back the dissemination of *X. fastidiosa* subsp. *multiplex* in France, various analyses were performed in this study. They required as input the use of genomic data that derived from genome sequences or microsatellite data from strains and/or from contaminated samples. Not all specimens of publicly available genome sequences were available at the time of this study. Likewise, the low bacterial population size of many French samples and the fact that they have been frozen since 2015 did not allow the isolation of strains for sequencing. In order to set up the most powerful analyses all the material available for each analysis was used, explaining why the number of samples differed between the analyses.

**Bacterial strains and growth conditions**. A collection of 95 *X. fastidiosa* strains (Supplementary Data 2) was used in this study. Among them, 44 come from the French Collection of Plant-Associated Bacteria (CIRM-CFBP. International Centre of Microbial Resource (CIRM) - French Collection for Plant-associated Bacteria. INRAE. https://doi.org/10.15454/E8XX-4Z18), Dr Leonardo De la Fuente (Auburn University, AL, USA) kindly provided 12 strains isolated in the USA, and the Plant Health Laboratory (Anses LSV, Angers, France) shared 39 strains isolated in France. All strains were preserved by the CIRM-CFBP. Strains were grown on BCYE medium[7] or modified PWG medium[54] at 28 °C for one to two weeks.

**Whole-genome sequencing**. Fifty-five strains were sequenced in this study (Supplementary Data 1). Forty-eight were sent to the BGI in Hong Kong for Illumina HiSeq X sequencing on HiSeq 4000 platform and seven strains were sent to the GeT-PlaGe core facility at the INRAE in Toulouse for PacBio sequencing. See supplementary material and methods for details on DNA extraction, quality check, library preparation and sequencing.

**Genome assembly, alignment, and SNP calling**. A total of 82 genome sequences of *X. fastidiosa* subsp. *multiplex* was analyzed in this study (Supplementary Data 1), including 27 publicly available genome sequences (NCBI on 16/12/2019) and the 55 obtained in this study. Genome assembly was performed with SOAPdenovo version 2.04, SOAPGapCloser version 1.12[55] and Velvet version 1.2.02[56]. The 82 genome sequences were aligned using the strain M12 as a reference with Parsnp tool v1.2 from Harvest suite[57] to obtain *X. fastidiosa* subsp. *multiplex* core genome. The matrix of SNPs was extracted using Gingr tool v1.3 from Harvest suite[57]. Detection of recombinant sequences within the core genome alignment was performed using ClonalFrame[58], running three independent MCMCs of 50,000 iterations, a burn-in length of 10,000 iterations and iteration samples every 100 iterations. To keep only SNPs due to mutations, the recombinant events detected were discarded from the SNP matrix using R scripts (M. Mariadassou and D. Merda pers. comm.) sourcing the "ape"[59] and "coda" packages[60].

**Population structure**. To infer population structure, the FastSTRUCTURE program based on a Bayesian clustering approach was run on the whole core SNPs alignment[61]. Ten independent runs were performed for $k = 1$ to 10, with a MCMC of 1,000,000 iterations. The other parameters were set as default. Then a heat map was drawn using the gplots package[62] of R software on SNPs non-recombinant, i.e., only issued from mutation events.

**Molecular tip dating**. The no-recombining core genome of the 82 strains, spanning 36 years of evolution (1983–2018) was used to investigate the presence of temporal signal thanks to two different tests. At first, a Maximum Likelihood (ML) tree was constructed with RAxML 8.2.4[63] using a rapid Bootstrap analysis, a General Time-Reversible model of evolution following a Γ distribution with four rate categories (GTRGAMMA) and 1000 alternative runs. A linear regression test between sample age and root-to-tip distances was computed at each internal node of the ML tree using Phylostems[64]. Temporal signal was considered present at nodes displaying a significant positive correlation. Secondly, a date-randomization test[65] was performed with 40 independent date-randomized datasets built with the R package "TipDatingBeast"[66]. The "RandomCluster" function allowed to randomized the isolation dates of samples per cluster of 10 years (1983–1989; 1990–1999; 2000–2009; 2010–2018). Results were generated using the "PlotDRT" function. Temporal signal was considered present when there was no overlap between both the inferred root height and substitution rate 95% HPD of the initial dataset and that of the date-randomized datasets. Tip-dating inferences were performed using the Bayesian MCMC method implemented in BEAST v2.6.1[36]. Leaf heights were constrained to be proportional to sample ages. Flat priors (i.e., uniform distributions) were applied both for the substitution rate ($10^{-12}$ to $10^{-2}$ substitutions/site/year) and for the age of all internal nodes in the tree. We also considered a general time reversible (GTR) substitution model with a Γ distribution and invariant sites (GTR + G + I), an uncorrelated relaxed log-normal clock to account for variations between lineages, and a tree prior for demography of coalescent Bayesian skyline. Six independent chains were run for 200,000,000 step and sampled every 20,000 steps, after a discarded burn-in of 10%. Convergence, sufficient sampling and mixing were verified by visual inspection of the posterior samples using Tracer v1.7.1[36]. Parameter estimation was based on the samples combined from the six different chains. The six trees output files were merged using LogCombiner software v2.6.1 after a 20% burn-in period and the best supported tree was drawn using the maximum clade credibility method implemented in Tree-Annotator software v2.6, after a 10% burn-in period[36]. In parallel, Bayesian skyline analysis was performed to observe effective population size along time using Tracer v1.7.1[36].

**Bacterial suspensions, DNA extraction and DNA sample collections used for MLVA**. For isolated strains (Supplementary Data 2), bacterial suspensions were

prepared from fresh cultures in sterile distilled water, adjusted at $OD_{600\,nm} = 0.1$ ($1 \times 10^8$ CFU.mL$^{-1}$), boiled for 20 min, cooled on ice and centrifuged at 10,000 g during 10 min. The 18 Spanish DNAs were kindly provided by Dr Blanca Landa (Institute for Sustainable Agriculture, Córdoba, Spain) (Supplementary Data 2). They were extracted using CTAB protocol[7].

The collection of 357 DNAs was obtained from infected plants collected in France (Corsica and PACA) between 2015 and 2018 in the framework of the national official surveillance strategy for *X. fastidiosa* and other dedicated sampling campaigns (Table 2, Supplementary Data 3). Samples were declared infected based on positive Harper's qPCR test at Cq<38[7]. Plant samples were finely chopped in distilled water, optionally sonicated[54], and incubated 15 min at room temperature before DNA extraction using CTAB or QuickPick™ SML Plant DNA Kit (Bio-Nobile, Turku, Finland) as described in PM7/24[7]. All the bacterial suspension and DNA samples were stored at −20 °C before analysis.

**VNTR-13 scheme genotyping**. The VNTR-13 scheme used in this study was composed of 10 VNTRs previously developed[37,38] and three newly-designed ones (Table 1). The VNTR amplification method was optimized for direct use on DNA extracted from infected plant material. See supplementary material and methods for details on optimization and final protocol.

**Data scoring**. Electropheregrams, obtained from capillary electrophoresis analysis of VNTRs, were analyzed using Geneious 9.1.8 software (Biomatters) and peaks were first automatically detected using the predict peaks mode. Each ladder and VNTR peaks were then carefully checked by eye and artefacts or small peaks were corrected. The 13 VNTRs amplicons were sequenced for the strains CFBP 8416, CFBP 8417, and CFBP 8418 to confirm length of the repeat flanking region, TR sequence length and its number of repeats. The number of repeats for each locus was calculated based on fragment size using Geneious 9.1.8 software (Biomatters).

**MLVA analyses and statistics**. MSTs were drawn and clonal complexes analyzed using the algorithm recommended for MLVA data combining global optimal Euclidean and goeBURST distances in PHYLOViZ v2.0[67]. Haplotypes differing one another by only one locus were grouped in clonal complexes. The discriminatory power of the MLVA was calculated using http://insilico.ehu.es/mini_tools/discriminatory_power/index.php. Data were analyzed using two different methods in order to infer population structure and assign samples to clusters. Bayesian clustering approach (STRUCTURE 2.3.4 software)[68] was applied with 10 independent runs, performed for $k = 1$ to 10, with a burn-in period of 100,000 iterations followed by 500,000 Monte Carlo Markov Chain (MCMC) replicates. This analysis was completed with the Discriminant Analysis of Principal Components (DAPC), which was used as it does not rely on any assumption about the population genetic model[69]. DAPC analyses were conducted on 20 independent k-means runs in order to verify the stability of the clustering. Analyses were performed using a single individual per haplotype with the adegenet package from the R software[70]. Simpson's index of diversity using BioNumerics 7.6 (Applied Maths) and allelic richness using Fstat 2.9.4[71] were calculated to access the discriminatory power of each VNTR. Using the groups defined by DAPC, the number of alleles and their frequency distribution were determined using GenAlEx 5.5.1[72]. Genetic differentiation of the DAPC groups was estimated using $F_{ST}$ and $R_{ST}$ with Arlequin 3.5.2.2[73]. Using the same software an analysis of molecular variance was performed. The individuals were grouped according to the groups established for DIYABC analysis. Spatial representation of the distribution of the French samples and their genetic clustering on the French territory were made using the R package maptools[74].

**Approximate Bayesian Computation (ABC) analysis**. We investigated whether the French emergence of *X. fastidiosa* subsp. *multiplex* was the result of multiple independent introduction events or the result of one or a few events that subsequently spread. First, we conducted a scenario choice using ABC. This method consists in generating a large number of datasets, simulated under each of the tested scenarios, and measuring the similarity between the simulated data and the real data with statistics[75]. The posterior probability of each scenario is then obtained from its occurrence in the simulations that are the closest to the real dataset using a post-sampling adjustment[75]. To decipher the scenario of introduction of *X. fastidiosa* subsp. *multiplex* in France, we first used an ABC approach which is implemented in the DIYABC software v2.1[76]. Then, the machine learning tool ABC random forest (ABC-rf), implemented in the R package "abcrf"[77] was used to analyzed DIYABC results, as it is the most robust and offers a larger discriminative power among competing scenarios[77,78].

Data were analyzed independently per sequence type number (ST6 and ST7). As *X. fastidiosa* subsp. *multiplex* ST6 and ST7 were previously known to occur only in the USA, the American ST6 Dixon strain and the American ST7 M12 and Griffin strains were used as representatives of ancestral populations, from which the French strains were linked. All MLVA data from French infected samples and strains were used to run ABC analyses. Four genetic groups were defined for ST6 and ST7 strains, using results previously obtained from DAPC and STRUCTURE clustering analysis, goeBURST tree topology, historical and geographical

information. For both ST analyses, three French populations were defined, one grouping both Corsica and PACA samples (named C1P1), one grouping exclusively Corsica samples (named C2) and one grouping exclusively PACA samples (named P2). The fourth population corresponded to the ancestral populations, composed of strains from USA. Based on these assumptions and data, the number of possible scenarios to test was estimated at 30, without taking into account the existence of unsampled population(s). In order to limit the number of scenarios analyzed in each run, a Nested strategy was followed. Populations were first compared by a bottom-up approach, in which scenarios were composed only of American strain(s) and two French populations. Then, the following analyses used the conclusions obtained from previous analyses and allowed to eliminate assumptions. Second, all French populations were simultaneously compared in a top-down approach, in which each scenario topology was analyzed separately. For a detailed description of the scenarios analyzed and software parameters read "Supplementary material".

**Reporting summary**. Further information on research design is available in the Nature Portfolio Reporting Summary linked to this article.

## Data availability

Strains of *X. fastidiosa* were deposited at the CIRM-CFBP (International Centre of Microbial Resource (CIRM) - French Collection for Plant-associated Bacteria. INRAE. https://doi.org/10.15454/E8XX-4Z18). Genome sequences were deposited at NCBI under the accession numbers listed in Supplementary Data 1. All data are available in the main text or the supplementary materials. Supplementary Data 8 contains the DiyABC analyses prior parameters.

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

## Acknowledgements

We thank Quentin Beaurepère for technical support, Muriel Bahut (ANAN technical facility, SFR QUASAV, Angers, FR) for DNA extraction automation and VNTRs sequencing, CIRM-CFBP (Beaucouzé, INRAE, France http://www6.inra.fr/cirm_eng/CFBP -Plant-Associated-Bacteria) for strain preservation and supply, Jérôme Gouzy (CATI BARIC) for the genome assembly pipeline, Mahendra Mariadassou and Déborah Merda for script sharing. We also thank Leonardo de la Fuente (Auburn University, AL, USA) and LSV-ANSES for sharing strains, Blanca L. Landa (Institute for Sustainable Agriculture, Córdoba, Spain) for sharing DNA; Valérie Oliver and Françoise Poliakoff for information related to the analysis of plant samples and strains in the framework of official surveillance, Marina Blanc for preliminary experiments on VNTR design, Laetitia Hugo from CNBC and François Casabianca from Corsican INRAE center for facilitating samplings, and Charles Manceau for co-writing the initial project providing E.D. funding support. INRAE SPE division and ANSES funded E.D. salary. European Union's Horizon 2020 research and innovation program under grant agreement 727987 XF-ACTORS (*Xylella fastidiosa* Active Containment Through a Multidisciplinary-Oriented Research Strategy) funded S.C. and M.A.J. The present work reflects only the authors' view and the EU funding agency is not responsible for any use that may be made of the information it contains. XyleCor funded by the SPE division of INRAE funded M.A.J. Sap Alien project funded by the RFI Objectif Végétal funded ND salary. France Génomique National infrastructure, funded as part of "Investissement d'avenir" program managed by Agence Nationale pour la Recherche (contract ANR-10-INBS-09) and by the GET-PACBIO program (« Programme operationnel FEDER-FSE MIDI-PYRENEES ET GARONNE 2014-2020 ») funded C.D. and C.L.R. European Union (European Regional Development Fund, ERDF contract GURDT I2016-1731-0006632), Conseil Régional De La Réunion and Centre de Coopération Internationale en Recherche Agronomique pour le Développement (CIRAD) funded A.R., O.P. and V.R.). Agence Nationale pour la Recherche JCJC MUSEOBACT contrat ANR-17-CE35-0009-01financed A.R.

## Author contributions

Conceptualization: M.A.J., E.D., S.C. Material and Methodology: E.D., S.C., M.A.J., V.R., A.R., K.D., M.B., O.P., N.D., C.D., C.L.R., A.C., B.L. Supervision: M.A.J., V.R., S.C., A.R. Writing-original draft: E.D., M.A.J., S.C., A.R., V.R. Writing-review & editing: E.D., M.A.J., V.R., A.R., O.P., M.B., K.D., S.C., A.C., B.L., N.D., C.D., C.L.R.

## Competing interests

The authors declare no competing interests.
