## [Peer Review File · Communications Biology]

Reviewers' comments:

Reviewer #1 (Remarks to the Author):

This manuscript represents a commendable effort to understand the emergence of *Xylella fastidiosa* in France. There are few other emerging plant pathogens that have received as much attention as the impacts of *X. fastidiosa* in Italy and there is a lot of concern about its potential impact in other regions. The authors use genome sequences to estimate the timing of diversification of strains and amplified fragment length polymorphisms (characterized VNTRs) to investigate epidemiological and ancestral relationships of strains. They uncover evidence of multiple introductions and hypothesize dispersal from two different outbreaks in southern France. The analysis is comprehensive and will be of great interest to those working on the epidemiology of highly regulated, emerging plant pathogens.

The main weakness of the paper is that the ABC analysis doesn't strongly support a single scenario or, really, a bridgehead. The authors downplay the results of the ABC analysis and emphasize early detections of the pathogen and knowledge of the distribution of a common host, *P. myrtifolia*, to support their argument of a southern France bridgehead. I think of a bridgehead as a major change in circumstances, often evolution or population size change, that allows multiple secondary invasions. What exactly the authors mean by a bridgehead should be clarified, since this is in the title of the paper. One could argue that the PACA introduction provided a "stepping-stone" to Corsica rather than a bridgehead.

Other comments

Of the 82 genomes, 27 were obtained from NCBI, including all the strains from Spain that were ST6 but genetically diverged. Were SRA files available for reassembly using the same pipeline as the ST6 strains from France?

Lines 296-316. The grouping of strains by DAPC is hard to follow. Consider presenting as Structure-style bar plots as then strains could be grouped and labeled by geographic origin. I'm also confused why STs were divided into different groups for the ABC than those defined by DAPC for $k=6$.

The TMRCA of ST7 from France and RAAR6Butte was estimated to be 1971 but ~1500 for M12 and Griffin, which were chosen to represent the USA population for the ABC analysis. I don't understand this decision. The introduction to France was not likely from the M12/Griffin population, but rather a population with shared ancestry with the USA strains. Also, is it typical to use only one strain (Dixon) to represent a population in ABC and this doesn't affect the summary statistics?

Minor comments

Line 72. "Transport of infected *Prunus* was the likely vector of the introduction of strains from the subspecies *multiplex* in southern Brazil" --- unclear where strains were introduced from and to, was Brazil the source or destination?

Line 131. Indicate in main text that this was clustering using fastStructure so the reader doesn't have to go to Fig. S1 to see the type of analysis used. This analysis used the SNPs after recombination was removed?

Consider ordering the methods the same as the results.

Fig. 1. The scale bar is measured in years?

Fig. S2. The quality of the heatmap is not good enough to read the values, the number of SNPs. Perhaps this is just the review quality, but should be improved in the supporting materials for publication.

Fig. S4. Typo in BEAST.

Fig. S11. I think scenarios are numbered differently in this figure than in Table S8.

Table S3. Why is this table separate from Table S2? And the loci in a different order?

There are a few instances of awkward phrasing.

Line 29. "emerging diseases, which frequency is increasing" instead use "emerging diseases, which are increasing in frequency"

Line 99. "Hence, situations seem having different histories" Unclear what is meant.

Line 273-275. I don't understand this sentence.

Reviewer #2 (Remarks to the Author):

The manuscript Two bridgehead invasions of *Xylella fastidiosa* subsp. *multiplex* in France reports on genetic characterization of samples of the plant pathogen *Xylella fastidiosa* using a combination of genomic data and copy number variants. The authors also conduct some analyses to test various introduction scenarios.

On the plus side, the authors have generated a lot of data to characterize the genetics of this important group of pathogens. On the minus side, the manuscript suffers from flaws in the description of the work done and lacks a clear goal and narrative. It almost feels as if the authors combined several studies into a single manuscript that they try to stitch together. It does not work well and the manuscript is confusing to read. The materials and methods are confusing and the description of the material used and/or obtained is difficult to understand; as described, it would be impossible to reproduce the work that was done.

At the beginning of the M&M the authors state that 95 XF strains were used in the study, then state that someone provided 12 more strains and someone else contributed 39 strains (are these strains part of the 95 or in addition to them?). In the following section they state that 18 strains were obtained from Spain. In the following section they are saying that 48 strains have been sequenced; then in the Genome Assembly section they state that 82 genomes were analyzed. Trying to figure out the math of all those numbers is like solving a puzzle!

One problem with this manuscript is that it is a combination of different types of analyses with different sets of samples, combined with the development of a new genotyping method. The development of the VNTR seems to be useful, in particular since it can be used directly with infected plant tissues, but it should probably be published elsewhere (or its development should be in the supp mat) and the validation of the markers properly validated and reported. It is not clear why this method was only developed for the multiplex complex. Was it because of the availability of the strains or of the genomes? Would the VNTR work with other subspecies, making it more applicable to other diseases on other crops?

The biggest problem with this manuscript is that it fails to directly address a biological question. It is very much

empirically driven and it is not clear what new knowledge is contributed. The entire "VNTR-13 scheme revealed that French *X. fastidiosa* split into four groups of strains" reads as the authors are just describing what they observed (how many alleles per locus, how many peaks in the electrophoregrams, how many clusters, etc). I am sure that the description is technically accurate and informative to those who study this pathogen, but I don't see how it can be of interest to a broader readership. The authors did attempt to resolve the invasion question with the ABC approach. The challenge is that they started with too many scenarios. They did make the effort to reduce the number of scenarios using a two step approach, but that still leaves 30 scenarios. With so many scenarios and so little support for each scenario (what can be concluded with <10% of the "votes"? If this had been an election, this scenario would have lost its seat!), we don't learn much about the invasion history from this analysis.

Overall, this manuscript contains some work that deserves to be published but would fail to be of interest to this journal with a broad readership and should be aimed at a more specialized journal, possibly in plant pathology or applied microbiology. The manuscript needs to be restructured so that the reader knows exactly what question is being addressed, what samples are used and what novel contribution it makes.

We thank the reviewers for their very useful comments and their help to improve our manuscript. Please find below our responses to each comment.

Reviewers' comments:

Reviewer #1 (Remarks to the Author):

This manuscript represents a commendable effort to understand the emergence of *Xylella fastidiosa* in France. There are few other emerging plant pathogens that have received as much attention as the impacts of *X. fastidiosa* in Italy and there is a lot of concern about its potential impact in other regions. The authors use genome sequences to estimate the timing of diversification of strains and amplified fragment length polymorphisms (characterized VNTRs) to investigate epidemiological and ancestral relationships of strains. They uncover evidence of multiple introductions and hypothesize dispersal from two different outbreaks in southern France. The analysis is comprehensive and will be of great interest to those working on the epidemiology of highly regulated, emerging plant pathogens.

→ We thank the Reviewer for this kind comment and interest in our study.

The main weakness of the paper is that the ABC analysis doesn't strongly support a single scenario or, really, a bridgehead. The authors downplay the results of the ABC analysis and emphasize early detections of the pathogen and knowledge of the distribution of a common host, *P. myrtifolia*, to support their argument of a southern France bridgehead. I think of a bridgehead as a major change in circumstances, often evolution or population size change, that allows multiple secondary invasions. What exactly the authors mean by a bridgehead should be clarified, since this is in the title of the paper. One could argue that the PACA introduction provided a "stepping-stone" to Corsica rather than a bridgehead.

→ We agree with the reviewer that what we meant by "bridgehead" should be better defined. We now provide the following literature definition. A "bridgehead" is "a concept [...] refers to widespread secondary invasions stemming from a particularly successful invasive population, which serves as a source of colonists for potentially remote new territories" (Estoup and Guillemaud, 2010) (lines 495-497). Here we demonstrate that Corsica was invaded at least twice (ST6 and ST7) by an already invasive population (PACA), which is in line with the idea of a bridgehead. Although we see the idea with the term stepping-stone, we do not think that it would clarify much what is going on in this emergence as the notion of stepping stone has been mainly much used in a context of established populations under a balance of evolutionary forces (e.g., "The model assumes that the entire population is subdivided into colonies and the migration of individuals in each generation is restricted to nearby colonies.", Kimura and Weiss, 1964). We have moderated our comments by changing the title to: Suspicions of two bridgehead invasions of *Xylella fastidiosa* subsp. *multiplex* in France

Other comments

Of the 82 genomes, 27 were obtained from NCBI, including all the strains from Spain that were ST6 but genetically diverged. Were SRA files available for reassembly using the same pipeline as the ST6 strains from France?

→ We used the assembly publicly available, as it is usually done in studies. We now provide this precision lines 531-538. As the genomes were obtained from different sequencing methods it is not possible to reassemble them all with a same pipeline. However, our results concerning the cleavage between French

and Spain strains are in line and confirmed the study of Landa et al., 2020, performed on only 2 French strains and 9 Spanish strains. The Landa's et al. study was the first one showing the polyphyly of ST6 based on samples from these two countries.

Lines 296-316. The grouping of strains by DAPC is hard to follow. Consider presenting as Structure-style bar plots as then strains could be grouped and labeled by geographic origin. I'm also confused why STs were divided into different groups for the ABC than those defined by DAPC for k=6.

→ We modified the text to make it easier to read on this part. As mentioned in the text, meaningful groups needed to be made in order to study the routes of invasion, using ABC analyses. At K=6, clonal complexes were divided into 2 or more groups, making no biological sense. So we chose to keep the K=4 clustering because *"this clustering made biological sense (clustering mainly coherent with geographical origin of the strain or potential dissemination of haplotypes)"* (L 274-278).

The TMRCA of ST7 from France and RAAR6Butte was estimated to be 1971 but ~1500 for M12 and Griffin, which were chosen to represent the USA population for the ABC analysis. I don't understand this decision. The introduction to France was not likely from the M12/Griffin population, but rather a population with shared ancestry with the USA strains. Also, is it typical to use only one strain (Dixon) to represent a population in ABC and this doesn't affect the summary statistics?

→ We agree with this comment and a part was added in the material and methods section to explain this point (lines 531-538). At the time of our study three ST7 and one ST6 genome sequences were publicly available. But the strain RAAR6_Butte (the closest to French ST7 strains) was not available, making it necessary for us to use both other ST7 strains. For ABC analyses an ancestral population is needed. The statistic would have been statistically stronger with more strains, but we performed the analyses with all the worldwide strains that were publicly available.

Minor comments

Line 72. "Transport of infected Prunus was the likely vector of the introduction of strains from the subspecies multiplex in southern Brazil" --- unclear where strains were introduced from and to, was Brazil the source or destination? *Brazil is the destination. (line 72-73, text modified to reflect this)*

Line 131. Indicate in main text that this was clustering using fastStructure so the reader doesn't have to go to Fig. S1 to see the type of analysis used. This analysis used the SNPs after recombination was removed?

→ All SNP analyses were performed after removal of recombination events (lines 131-132, text modified to reflect this)

Consider ordering the methods the same as the results. *(text modified to reflect this)*

Fig. 1. The scale bar is measured in years? *Yes, it is (legend modified to reflect this)*

Fig. S2. The quality of the heatmap is not good enough to read the values, the number of SNPs. Perhaps this is just the review quality, but should be improved in the supporting materials for publication. *The quality was improved. If the visual is not better in the review and proof, we will contact the editor to find a way to upload it individually.*

Fig. S4. Typo in BEAST. *(text modified to reflect this)*

Fig. S11. I think scenarios are numbered differently in this figure than in Table S8. (text modified to reflect this)

Table S3. Why is this table separate from Table S2? And the loci in a different order? The table S2 grouped data obtained from VNTR amplification from boiled strains, while table S3 data were obtained from extracted DNA. We agree that this difference is subtle, so the two tables have been fused and this difference added as a legend.

There are a few instances of awkward phrasing. All these sentences have been modified

Line 29. “emerging diseases, which frequency is increasing” instead use “emerging diseases, which are increasing in frequency”

Line 99. “Hence, situations seem having different histories” Unclear what is meant.

Line 273-275. I don't understand this sentence.

Reviewer #2 (Remarks to the Author):

The manuscript Two bridgehead invasions of *Xylella fastidiosa* subsp. *multiplex* in France reports on genetic characterization of samples of the plant pathogen *Xylella fastidiosa* using a combination of genomic data and copy number variants. The authors also conduct some analyses to test various introduction scenarios.

On the plus side, the authors have generated a lot of data to characterize the genetics of this important group of pathogens. On the minus side, the manuscript suffers from flaws in the description of the work done and lacks a clear goal and narrative. It almost feels as if the authors combined several studies into a single manuscript that they try to stitch together. It does not work well and the manuscript is confusing to read. The materials and methods are confusing and the description of the material used and/or obtained is difficult to understand; as described, it would be impossible to reproduce the work that was done.

We agree with reviewer #2 that different analyses were used in this study and that they were based on different materials and types of analysis. Nevertheless, they are all necessary to decipher the history of the introduction of *Xylella* in France, as it is not possible to date the divergence and estimate routes of invasion with a single analysis. We have improved the narrative in order to be clearer and more precise.

At the beginning of the M&M the authors state that 95 XF strains were used in the study, then state that someone provided 12 more strains and someone else contributed 39 strains (are these strains part of the 95 or in addition to them?). In the following section they state that 18 strains were obtained from Spain. In the following section they are saying that 48 strains have been sequenced; then in the Genome Assembly section they state that 82 genomes were analyzed. Trying to figure out the math of all those numbers is like solving a puzzle! One problem with this manuscript is that it is a combination of different types of analyses with different sets of samples, combined with the development of a new genotyping method. The development of the VNTR seems to be useful, in particular since it can be used directly with infected plant tissues, but it should probably be published elsewhere (or its development should be in the supp mat) and the validation of the markers properly validated and reported.

→ We thank reviewer #2 for this comment and as he/she suggested the development of the MLVA scheme (material and methods + results) was transferred in a supplemental part. Nevertheless, the validation of the markers was properly done and reported, as validation was performed: i) *in silico* on genomic data, ii) *in vitro* on strains, extracted DNA and extracted DNA of infected samples, iii) specificity was validated on the Whole Genome Shotgun database of the NCBI (as on August 22, 2018), iv) comparison between results obtained on boiled strains and extracted DNA from plant validated the use of the scheme on this kind of singular material, iiv) description of the alleles range, allelic diversity and richness shown.

It is not clear why this method was only developed for the multiplex complex. Was it because of the availability of the strains or of the genomes?

→ The VNTR13 scheme was developed to be the most discriminant possible on strains of the subspecies *multiplex*, as this subspecies is the only one present in France, excepted a few foci previously reported that were eradicated (Denancé et al., 2017; Cuntly et al., 2022). However, the possible amplification of all VNTRs in strains of the other subspecies of *X. fastidiosa* was validated *in silico* and *in vitro* to indicate the general value of our scheme. It is mentioned in the Material and Methods section (L616-619 old version, supplementary materials L40-44 in the new version) “The specificity of all 13 VNTR primer pairs was tested *in silico* using PrimerSearch (Val Curwen, Human Genome Mapping Project, Cambridge, UK) on the 154,478 bacterial Whole Genome Shotgun (WGS) sequences available in the NCBI database (as on August 22, 2018). The primer pair sets were tested using Amplify to verify the absence of dimer and cross-amplification with other bacterium species.” and in the Results section (L183-189 and 193-198 old version, supplementary materials L66-79 in the new version) “These 13 VNTR loci were present in all *X. fastidiosa* genome sequences examined” [...] “*in silico* analyses on the WGS database of bacteria from NCBI revealed that specific amplifications were exclusively obtained on all *X. fastidiosa* genome sequences and on no other genome sequence” “The VNTR-13 scheme was specifically developed to type the subspecies *multiplex*. [...] Interestingly, each VNTR locus was also successfully amplified on the 27 DNAs of *X. fastidiosa* strains belonging to nine different sequence types of *X. fastidiosa* subsp. *fastidiosa* and *pauca* (Table S2, Fig. S5).”

Would the VNTR work with other subspecies, making it more applicable to other diseases on other crops?

→ Indeed, as mentioned above and in the manuscript, the possible amplification of all VNTRs in strains of the other subspecies of *X. fastidiosa* was validated *in silico* and *in vitro* to indicate the general value of our scheme (Please refer to Lines 66-67 and 79-81 in the supplemental materials).

The biggest problem with this manuscript is that it fails to directly address a biological question. It is very much empirically driven and it is not clear what new knowledge is contributed. The entire “VNTR-13 scheme revealed that French *X. fastidiosa* split into four groups of strains” reads as the authors are just describing what they observed (how many alleles per locus, how many peaks in the electrophoregrams, how many clusters, etc). I am sure that the description is technically accurate and informative to those who study this pathogen, but I don't see how it can be of interest to a broader readership. The authors did attempt to resolve the invasion question with the ABC approach. The challenge is that they started with too many scenarios. They did make the effort to reduce the number of scenarios using a two-step approach, but that still leaves 30 scenarios. With so many scenarios and so little support for each scenario (what can be concluded with <10% of the “votes”? If this had been an election, this scenario would have lost its seat!), we don't learn much about the invasion history from this analysis.

→ In DiyABC, it is essential not to constrain the scenarios. As we have no data allowing us to propose a priori a preferred area of introduction or a preferred direction of dissemination, we have chosen to do a 2-step analysis. This choice is therefore totally linked to the context of the study.

The number of scenarios is high for ST6 (30 scenarios) because the statistics did not allow us to eliminate more scenario during the first approach (bottom-up), in contrast to what we were able to do for ST7, which allowed us to keep only 12 scenarios for the second step.

The percentage of votes are low. This is why we tempered our words by also looking at the percentage of votes validating the divergence of the ST6_C2 population from the ST6_P2 one (36%) and the introduction of the ST7_P2 population from which would have diverge the two other populations (53%). We also insisted a lot on the exclusion of scenario that received very low percentage of votes.

We could have tried to ran an analysis selecting only the 5 best scenario from our analysis, but this is clearly not recommended as it would have meant to compare 4 scenarios including a first introduction in PACA (II.7, II.9, IV.25 and IV.27) vs only one scenario including a first introduction in Corsica (II.11). This would have inevitably induced the dilution of the votes for the first introduction in PACA. If we use the referee's illustration again, it would have meant presenting 4 candidates from one party and only one for the opposition, which clearly favor the opposition due to vote dilution.

Overall, this manuscript contains some work that deserves to be published but would fail to be of interest to this journal with a broad readership and should be aimed at a more specialized journal, possibly in plant pathology or applied microbiology. The manuscript needs to be restructured so that the reader knows exactly what question is being addressed, what samples are used and what novel contribution it makes.

→ Future sampling and statistical analyses could further improve and hopefully validate our proposed scenario. However, to date, very few articles deal with the emergence of bacterial disease based on so many data and an attempt to trace the routes of invasion, making it a study of great interest. Moreover, the dating of the introduction in France is statistically reliable. In addition, we have developed a MLVA scheme that could be used worldwide on the different subspecies of *X. fastidiosa*. This scheme highlighted a high diversity between the samples and validated an ancient introduction. It is also the first and only study analyzing the diversity of the entire collection of French samples collected between 2015 and 2018, other than by MLST, a much less resolving method, whose only information was the differentiation of strains in two groups: ST6 or ST7. To date the study of *Xylella* routes of invasion has only been performed on 45 genome sequences in the study of Landa *et al.*, 2020 study and by tip-dating in the study of Moralejo *et al.*, 2020 on 15 Majorcan strains, a study that was published in the Journal Communications biology. As this is the first study showing a bridgehead invasion in France and as many studies, all pathogens considered, try to trace back and date invasion routes, we think that our study and its approach should be of interest for the large readership of this generalist journal.

Reviewers' comments:

Reviewer #1 (Remarks to the Author):

The authors largely addressed my comments from their previous submission. Historical reconstruction of disease emergence is challenging. Despite the limitations identified in the previous reviews, I believe that this analysis is important and impactful.

There are a few remaining minor issues in the manuscript.

1. The abbreviation PACA is used in the abstract, but I don't think this abbreviation will be widely known.
2. line 248, I believe "ought" should be "sought"
3. The heat map in Fig. S2 is still illegible. It needs to be imported into the word document in higher resolution or figures should be assembled into a pdf directly, bypassing word.

Reviewer #2 (Remarks to the Author):

This manuscript describes a population study of a plant pathogenic bacteria in southern Europe. The study is probably better suited to a plant pathology journal as it is very descriptive and I do not believe that it makes a significant contribution to our understanding of plant disease outbreaks or invasions of plant pathogens and is not suited to a journal with a broad readership such as Comms Biol. The focus on two regions of France and two strains of the multiplex subspecies (ST6 and ST7) is not well explained to the reader who does know about this disease. Why is it important in that region and those strains? Is there a biological reason or a regulatory one? The introduction is long and lacks focus, providing information that is not essential to understand what is being proposed in this work. The aim of the work seems to be to assess the genetic relatedness between strains originating from Corsica and PACA. The new method developed could be interesting but it is not clear if the author's claim of "Better genotyping resolution with MLVA within *X. fastidiosa* subsp multiplex" is supported. Better than what? How much better, and what level of resolution is really required. Ideally, the development of this new method would comprise a proper validation and assessment of level of improvement. The manuscript seems to be much longer than the proposed 5000 words from the journal. I believe that shortening both the introduction and discussion would clarify what was done and why it is important and have a higher impact. There are also multiple typos and grammatical errors that would need to be fixed prior to publication. My suggestion would be to shorten the paper and focus the narrative on the major findings.

Reviewer #1 (Remarks to the Author):

The authors largely addressed my comments from their previous submission. Historical reconstruction of disease emergence is challenging. Despite the limitations identified in the previous reviews, I believe that this analysis is important and impactful.

→ We thank the Reviewer for the support of our work.

There are a few remaining minor issues in the manuscript.

1. The abbreviation PACA is used in the abstract, but I don't think this abbreviation will be widely known.
→ text modified to reflect this
2. line 248, I believe "ought" should be "sought"
→ text modified to reflect this
3. The heat map in Fig. S2 is still illegible. It needs to be imported into the word document in higher resolution or figures should be assembled into a pdf directly, bypassing word.
→ we will upload it in power-point format, which should correct the problem.

Reviewer #2 (Remarks to the Author):

This manuscript describes a population study of a plant pathogenic bacteria in southern Europe. The study is probably better suited to a plant pathology journal as it is very descriptive and I do not believe that it makes a significant contribution to our understanding of plant disease outbreaks or invasions of plant pathogens and is not suited to a journal with a broad readership such as Comms Biol.

The focus on two regions of France and two strains of the multiplex subspecies (ST6 and ST7) is not well explained to the reader who does know about this disease. Why is it important in that region and those strains? Is there a biological reason or a regulatory one?

→ We rephrase L76 to 80

For the samples we have access to, i.e. those collected between 2015 and 2018, these two regions were the only ones in France that were infected with *X. fastidiosa*. At the exception of only 2 foci, all infections were caused by *X. fastidiosa* ST6 or ST7.

The introduction is long and lacks focus, providing information that is not essential to understand what is being proposed in this work.

→ We thank reviewer 2 for this comment. We reduced and reordered the introduction to be shorter and more focus on our subject.

The aim of the work seems to be to assess the genetic relatedness between strains originating from Corsica and PACA. The new method developed could be interesting but it is not clear if the author's claim of "Better genotyping resolution with MLVA within *X. fastidiosa* subsp multiplex" is supported. Better than what? How much better, and what level of resolution is really required.

→ MLST, which is the reference method to type *Xylella* worldwide, types 99% of the French strains into only two lineages, ST6 or ST7, whereas our MLVA scheme typed them into 320 haplotypes. Within each lineage defined by MLST, the strains are not distinguishable, and no further analysis were possible.

Moreover, using the http://insilico.ehu.es/mini_tools/discriminatory_power/index.php the discriminatory power of the MLST on the 396 samples is of 0.4976 (data not shown in the article), when the discriminatory power of the MLVA is of 0.9969 (L182)

Ideally, the development of this new method would comprise a proper validation and assessment of level of improvement.

→ Given that the method was developed and tested i) in silico on genomic data, ii) in vitro on ranges of strains from the different subspecies, then iii) on naturally infected samples and that iv) some TRs were sequenced to verify their lengths and compositions, we do not understand why Reviewer 2 does not consider this to be a “proper validation”. We also provide rarefaction curves to illustrate the appropriateness of this scheme for our data. We are not aware of any additional step required to validate the development of a VNTR scheme.

The manuscript seems to be much longer than the proposed 5000 words from the journal.

→ We agreed that it is a little bit longer than usual, because we developed a new method and used multidisciplinary methods. In this new version we made effort to reduce by 1800 words the length of our manuscript. We choose to present all the methods and explain precisely protocols and results so that our work could be reproduced or reused in other context. We also transferred in the supplementary material most parts of the VNTR development to me more focused on divergence dating and reconstruction of invasion scenario. Indeed, this transfer considerably lightens the Results section. We shall indicate here that we prefer to keep the development of this MLVA scheme attached to this MS and do not wish to publish it separately, as it is closely related to its use directly on plant material to trace the introduction and dissemination routes of Xf, the core part of this MS.

I believe that shortening both the introduction and discussion would clarify what was done and why it is important and have a higher impact. There are also multiple typos and grammatical errors that would need to be fixed prior to publication.

→ We followed this advice and corrected typos and grammatical errors we saw.

My suggestion would be to shorten the paper and focus the narrative on the major findings.

→ We thank Reviewer 2 for this suggestion that we took on.